# Management and outcomes of patients with chronic obstructive lung disease and lung cancer in a public healthcare system

**John R. Goffin**[1]*, **Sophie Corriveau**[2], **Grace H. Tang**[3], **Gregory R. Pond**[1]

1 Department of Oncology, McMaster University, Hamilton, Ontario, Canada, 2 Division of Respirology, Department of Medicine, McMaster University, Hamilton, Ontario, Canada, 3 Institute of Health Policy, Management, and Evaluation, University of Toronto, Toronto, Ontario, Canada

* goffin@mcmaster.ca

**Data Availability Statement:** The data acquired for this work was obtained under ICES Date Use Agreement # 2015-066. The authors did not have privileged access to this data. Data may be

## Abstract

### Hypothesis

There is limited data on the care and outcomes of individuals with both chronic obstructive pulmonary disease (COPD) and lung cancer, particularly in advanced disease. We hypothesized such patients would receive less cancer treatment and have worse outcomes.

### Methods

We analyzed administrative data from the province of Ontario including demographics, hospitalization records, physician billings, cancer diagnosis, and treatments. COPD was defined using the ICES-derived COPD cohort (1996–2014) with data from 2002 to 2014. Descriptive statistics and multivariable analyses were undertaken.

### Results

Of 105 304 individuals with lung cancer, 43 375 (41%) had stage data and 36 738 (34.9%) had COPD. Those with COPD were likely to be younger, have a Charlson score ≤ 1, have lower income, to live rurally, and to have stage I/II lung cancer (29.8 vs 26.5%; all p<0.001). For the COPD population with stage I/II cancer, surgery and adjuvant chemotherapy were less likely (56.8 vs. 65.9% and 15.4 vs. 17.1%, respectively), while radiation was more likely (26.0 vs 21.8%) (p all < 0.001). In the stage III/IV population, individuals with COPD received less chemotherapy (55.9 vs 64.4%) or radiation (42.5 vs 47.5%; all p<0.001). Inhaler and oxygen use was higher those with COPD, as were hospitalizations for respiratory infections and COPD exacerbations. On multivariable analysis, overall survival was worse among those with COPD (HR 1.20, 95% CI 1.19–1.22).

### Conclusions

A co-diagnosis of COPD and lung cancer is associated with less curative treatment in early stage disease, less palliative treatment in late stage disease, and poorer outcomes.

obtained from the third party and are not publicly available. A data request can be sent to ICES (formerly the Institute for Clinical Evaluative Sciences): https://www.ices.on.ca/About-ICES/ICES-Contacts-and-Sites/contact-form.

**Funding:** The authors received no direct funding for this work. Support to ICES is provided from an annual grant by the Ministry of Health and Long-Term Care (MOHLTC) and the Ontario Institute for Cancer Research (OICR). The opinions, results and conclusions reported in this paper are those of the authors. No endorsement by ICES, Cancer Care Ontario, OICR or the Government of Ontario is intended or should be inferred. The funding source had no role in the collection, analysis, and interpretation of data and in writing the manuscript.

**Competing interests:** Dr Goffin declares receiving Honorarium from Eisai (2020), Bristol-Myers Squibb (2020) and Merck (2018), conference travel support from AstraZeneca (2017), a speaking fee from Amgen (2018), and funding from the Canadian Partnership Against Cancer. All other authors report no conflict of interest. This does not alter our adherence to PLOS ONE policies on sharing data and materials.

# Introduction

Chronic obstructive pulmonary disease (COPD) has an estimated global prevalence of 10.7% in the population over 30, while lung cancer is both the most commonly diagnosed cancer and the leading cause of cancer death globally [1, 2]. The presence of COPD is associated with increased risk for the development of lung cancer (hazard ratio 2.2) adjusting for smoking history [3]. It is also evident that testing for COPD by spirometry is underutilized, including in the lung cancer population [4, 5]. Co-incident diagnoses of COPD and lung cancer may have implications for management of lung cancer and patient outcomes.

COPD itself is associated with higher mortality in the general population, with higher Global Initiative on Obstructive Lung Disease (GOLD) scores portending worse prognosis [6]. In the general cancer population, worsening COPD also appears to contribute to higher mortality [7]. In a large meta-analysis of studies from the USA, Europe, and Asia, Gao et al found that survival was worse among patients with COPD in addition to lung cancer; greater effect was seen in the stage I-II population, while data limits disallowed specific analysis of the stage III-IV population [8]. Two smaller studies did not find worse survival with COPD on multivariable analysis in the stage III-IV population [9, 10].

Some studies suggest that the presence of COPD alters treatment received for lung cancer, with fewer patients undergoing surgery and more complications occurring in those who do [11–13]. Conversely, data on the use of chemotherapy and radiotherapy among those with COPD is less clear [11, 12]. Patients with either lung cancer or COPD have high rates of hospital contact within the last 6–12 months of life [14, 15]. Treatment of COPD has been shown to decrease hospitalization and improve quality of life while being cost effective, and appropriate management may improve both symptom burden and healthcare resource consumption [16–19].

With the high prevalence of COPD and lung cancer, as well as improved treatments for lung cancer conferring longer survival, the relevance of COPD to lung cancer management and outcomes is likely to grow [20]. The present study therefore addresses questions of management and disease outcomes of patients with both COPD and lung cancer in a large population-based administrative database within a public healthcare system. We hypothesized that a dual diagnosis would decrease the rates of surgery, radiation, and systemic therapy provided and increase the use of supportive medications. In addition, the study addressed the hypothesis that patients with COPD would be more commonly hospitalized for respiratory infections and would have worse overall mortality.

# Materials and methods

Anonymized data was collated by ICES (previously Institute for Clinical Evaluative Sciences), which links administrative data from several databases for all patients in the province of Ontario in Canada. Analysis was done using ICES's confidential, analytic, virtual environment with SAS v9.3.

COPD was defined through the ICES-derived COPD cohort from 1996 to 2014. This ICES cohort is derived from hospital records and physician billing data, and was originally validated through comparison to a primary care chart review, achieving a sensitivity of 85% and specificity of 78% for defining COPD for a population age 35 years and older [21]. Data from 1994 to 2014 was collected for cancer diagnosis and stage through the Ontario Cancer Registry. The present cohort was limited to the population diagnosed with lung cancer from 2002 to 2014, allowing a period for capture of COPD diagnosis prior to a diagnosis of lung cancer. Demographic information including age, sex, vital status, income quintile, and rural status (rural defined as community size less than 10 000) were derived from the provincial health plan's

Registered Persons Database and Postal Code Conversion File. Comorbidity, emergency department use, and hospitalization were derived from the National Ambulatory Care Reporting System, the Discharge Abstract Database and the Ontario Hospital Insurance Plan for physician billing. The Adjusted Clinic Group Resource Utilization Band (ACG RUB) uses demographics and medical diagnoses to score 6 levels of risk for requiring further medical resources. Charlson score and ACG RUB are defined using 2 years of administrative date prior to the first diagnosis of COPD or lung cancer. Drug treatment was determined through the New Drug Funding Program for chemotherapy (available for all ages) and through the Ontario Drug Benefit claims for inhaler use (available for those at least years of 65 and those with disability benefits). Oxygen use was determined through the Assistive Devices program. Early stage (I/II) and late stage (III/IV) cancer treatments were designed to be independent, with treatment after early stage treatment being limited to surgery within 3 months of diagnosis or chemotherapy or radiation initiated within 6 months.

Descriptive statistics were used throughout. Comparisons between two groups of patients was undertaken using Wilcoxon rank sum tests (for continuous outcomes) and chi-square tests (categorical outcomes). Cox proportional hazards regression was used to examine factors potentially prognostic of overall survival. Logistic regression analysis was used to examine factors potentially prognostic of different types of treatment. All tests were two-sided and a p-value of 0.05 or less was considered statistically significant. In this exploratory analysis, no adjustment was made for multiple testing. However, appropriate caution was made when making inferences from the study results.

The study was approved by the Hamilton Integrated Research Ethics Board.

## Results

From 1996 to 2014, 146 787 individuals were diagnosed with lung cancer in Ontario. We excluded 41 314 diagnosed from 1996–2001, 12 with no follow-up after diagnosis, and 157 with death or last follow-up dated prior to cancer diagnosis (indicating data error or diagnosis at autopsy). After exclusions, 105 304 individuals were diagnosed with lung cancer from 2002 to 2014 (Table 1). Among these individuals, 43 375 had stage data (41%). The group having COPD, representing 34.9% of the lung cancer population, was younger, was less likely to have a Charlson score of at least one, was less likely to be attributed a higher Resource Utilization Band, was somewhat more likely to have lower income and live in a rural location, and was more likely to have stage I/II disease (29.8 vs. 26.5%) (all p<0.001).

Among persons with stage I/II disease, surgery within 6 months of diagnosis of lung cancer was less likely among individuals with COPD (56.8 vs 65.9%) (Table 2). Conversely, the COPD group was more likely to receive radiation within 6 months of diagnosis (26.0 vs 21.8%). Adjuvant chemotherapy (that provided within 6 months of diagnosis) was uncommon in both groups, but less likely to be received by the COPD group (15.4 vs 17.1%) (p all < 0.001). The group with COPD was more likely to be treated with oxygen or any inhaler and was more likely to suffer respiratory infection or a COPD exacerbation requiring hospital contact or hospitalization. Survival was superior in the group without COPD, with a median of 70.8 months (95% CI 66.5–75.2) versus 46.2 months (95% CI 43.8–48.0) (p < 0.001).

In the population with stage III/IV disease, individuals with COPD were more likely to never receive chemotherapy (64.4 vs. 55.9%) (p < 0.001) and were also less likely to receive an oral tyrosine kinase inhibitor (TKI) (2.5 vs 4.7%) (p < 0.001) (among individuals eligible for public TKI funding) (Table 3). Radiation was less commonly provided to the COPD population (never provided in 47.5 vs. 42.5%) (p < 0.001). In terms of supportive treatment, individuals with COPD were more likely to receive any inhaler (62.5 vs 33.3%) (p < 0.001) and to

**Table 1. Population demographics.**

| Characteristic | | N | All Patients | Prior COPD | No Prior COPD |
|---|---|---|---|---|---|
| N | | | 105304 | 36738 | 68566 |
| Sex | N (%) Female | 105304 | 49453 (47.0) | 17127 (46.6) | 32326 (47.2) |
| Age Group | ≤39 | 105304 | 882 (0.8) | 235 (0.6) | 647 (0.9)* |
| | 40–44 | | 1594 (1.5) | 673 (1.8) | 921 (1.3) |
| | 45–49 | | 3746 (3.6) | 1563 (4.3) | 2183 (3.2) |
| | 50–54 | | 6973 (6.6) | 2793 (7.6) | 4180 (6.1) |
| | 55–59 | | 11099 (10.5) | 4574 (12.5) | 6525 (9.5) |
| | 60–64 | | 14907 (14.2) | 5999 (16.3) | 8908 (13.0) |
| | 65–69 | | 17347 (16.5) | 6985 (19.0) | 10362 (15.1) |
| | 70–74 | | 17213 (16.4) | 6152 (16.8) | 11061 (16.1) |
| | 75–79 | | 14959 (14.2) | 4419 (12.0) | 10540 (15.4) |
| | 80–84 | | 10111 (9.6) | 2301 (6.3) | 7810 (11.4) |
| | 85+ | | 6473 (6.2) | 1044 (2.8) | 5429 (7.9) |
| Charlson Score | N (%) ≥1 | 105304 | 13503 (12.8) | 3415 (9.3) | 10088 (14.7)* |
| Resource Utilization Band | 0 | 105304 | 2186 (2.1) | 952 (2.6) | 1234 (1.8)* |
| | 1 | | 1717 (1.6) | 940 (2.6) | 777 (1.1) |
| | 2 | | 5150 (4.9) | 2674 (7.3) | 2476 (3.6) |
| | 3 | | 45558 (43.3) | 18727 (51.0) | 26831 (39.1) |
| | 4 | | 28314 (26.9) | 8269 (22.5) | 20045 (29.2) |
| | 5 | | 22379 (21.3) | 5176 (14.1) | 17203 (25.1) |
| Income Quintile | N (%) Lowest | 104843 | 24522 (23.4) | 9131 (25.0) | 15391 (22.6)* |
| | 2 | | 23327 (22.3) | 8493 (23.2) | 14834 (21.7) |
| | 3 | | 20506 (19.6) | 7199 (19.7) | 13307 (19.5) |
| | 4 | | 19211 (18.3) | 6258 (17.1) | 12953 (19.0) |
| | Highest | | 17277 (16.5) | 5520 (15.1) | 11757 (17.2) |
| Rural | N (%) Yes | 105189 | 17188 (16.3) | 6710 (18.2) | 10593 (15.3)* |
| Stage | N (%) 1 | 43375 | 8639 (19.9) | 3607 (21.2) | 5032 (19.1)* |
| | 2 | | 3422 (7.9) | 1464 (8.6) | 1958 (7.4) |
| | 3 | | 8807 (20.3) | 3717 (21.9) | 5090 (19.3) |
| | 4 | | 22507 (51.9) | 8205 (48.3) | 14302 (54.2) |

receive home oxygen (21.7 vs 15.7%) (p < 0.001). The stage III/IV population with COPD was also more likely to be seen in the emergency department or hospitalized with pneumonia (46.4 vs 31.2%) (p < 0.001) or a COPD exacerbation (30.4% vs 6.9%) (p < 0.001) during their course of care. Overall survival was inferior in the COPD group, with median survival of 4.3 months (95% CI 4.1–4.5) versus 5.3 months (95% CI 5.2–5.5) for those without COPD, and 1-year survival of 26.3% (95% CI 25.5–27.1) versus 29.8% (95% CI 29.1–30.4) (p< 0.001).

On multivariable analysis, in addition to greater age, male sex, being attributed a lower Resource Utilization Band, having a Charlson score at least 1, having lower income, having a rural residence, and having a higher stage, the presence of COPD independently decreased the likelihood of undergoing surgery within six months of lung cancer diagnosis (Odds Ratio 0.85, 95% CI 0.81–0.88) (Table 4). Similar factors influenced the likelihood of receiving chemotherapy, with the notable difference that chemotherapy was more likely to be provided as years progressed over the period of the study (per year Odds Ratio 1.03, 95% CI 1.02–1.03), while surgery was less likely (Odds Ratio 0.96, 95% CI 0.96–0.97). On multivariable analysis, the presence of COPD independently decreased the likelihood of receiving chemotherapy (HR 0.60, 95% CI 0.58–0.64).

**Table 2. Management and outcomes of stage I/II patients.**

| Characteristic | | No COPD on Day Lung CA diagnosed (N = 6990) | COPD on Day Lung CA diagnosed (N = 5071) | P value |
|---|---|---|---|---|
| Lung Surgery | N (%) ≤30-day pre to 6 Months Post-Dx | 4604 (65.9) | 2881 (56.8) | <0.001 |
| 1st Chemotherapy | Never | 5203 (74.4) | 3928 (77.5) | <0.001 |
| | <6 months Post Lung Dx | 1194 (17.1) | 782 (15.4) | |
| | >6 months Post Lung Dx | 593 (8.5) | 361 (7.1) | |
| Any Oral Anti-Cancer Agent | N (%) Yes | 112 (1.6) | 54 (1.1) | 0.012 |
| 1st Radiation | Never | 4432 (63.4) | 3000 (59.2) | <0.001 |
| | <6 months Post Lung Dx | 1523 (21.8) | 1320 (26.0) | |
| | >6 months Post Lung Dx | 1035 (14.8) | 751 (14.8) | |
| Oxygen | N (%) Yes | 529 (7.6) | 795 (15.7) | <0.001 |
| Short Acting Beta Agonist | N (%) Yes | 2503 (35.8) | 2997 (59.1) | <0.001 |
| Long Acting Beta Agonist | N (%) Yes | 1699 (24.3) | 2525 (49.8) | <0.001 |
| Short Acting Anticholinergic | N (%) Yes | 640 (9.2) | 963 (19.0) | <0.001 |
| Long Acting Anticholingeric | N (%) Yes | 1629 (23.3) | 2744 (54.1) | <0.001 |
| Inhaled Corticosteroid | N (%) Yes | 937 (13.4) | 1447 (28.5) | <0.001 |
| Any Inhaler | N (%) Yes | 3199 (45.8) | 3595 (70.9) | <0.001 |
| Influenza | N (%) Yes | 80 (1.1) | 99 (2.0) | <0.001 |
| Pneumonia | N (%) Yes | 1661 (23.8) | 2114 (41.7) | <0.001 |
| COPD exacerbation | n/N (%) Yes | 722 (10.3) | 1805 (35.6) | <0.001 |
| Overall Survival | N (%) Deaths | 2629 (37.6) | 2405 (47.4) | <0.001 |
| | Median (95% CI) Months | 70.8 (66.5, 75.2) | 46.2 (43.8, 48.0) | |
| | 1-year (95% CI) OS | 85.8% (85.0, 86.6) | 80.1% (78.9, 81.1) | |

Overall survival was worse with older age (HR 1.02 /5-year interval, 95% CI 1.02–1.02), a Charlson score of at least 1 (HR 1.22, 95% CI 1.20–1.25), and higher stage, and was better among women (HR 0.85, 95% CI 0.84–0.86), higher Resource Band allocation (HR 0.95 per unit, 95% Ci 0.94–0.96), higher income (HR 0.97 per quintile, 95% CI 0.96–0.97), and urban habitation (HR 0.95, 95% CI 0.94–0.96). The presence of COPD was independently associated with worse survival (HR 1.20, 95% CI 1.19–1.22) (Table 4).

## Discussion

In this large, population-based data set from a public healthcare system, we found that survival was inferior among individuals having a concomitant diagnosis of COPD and lung cancer, independent of demographics, comorbidity, and stage. Our hazard ratio is very similar to the value found by Gao et al in their meta-analysis (HR 1.20 in our dataset vs 1.17) [8]. While Gao et al had insufficient data to comment on the effect of survival beyond early stage disease, our data shows a significantly worse survival among the COPD population in both early and late stage disease. Smaller studies (n = 324 and n = 337) did not find differences in survival in the population with both advanced lung cancer and COPD, although sample size may have been a factor [9, 10].

The reasons for worse survival among patients with COPD may be several. In early stage patients, we found individuals with COPD were less likely to get surgery, the gold standard for cure. Persons with COPD did receive more radiation, possibly as an alternative to surgery,

**Table 3. Management and outcomes of stage III/IV patients.**

| Characteristic | | No COPD on Day Lung CA diagnosed (N = 19392) | COPD on Day Lung CA diagnosed (N = 11922) | P value |
|---|---|---|---|---|
| Lung Surgery | N (%) ≤30-day pre to 6 Months Post-Dx | 1086 (5.6) | 612 (5.1) | 0.077 |
| 1st Chemotherapy | Never | 11032 (55.9) | 7674 (64.4) | <0.001 |
| | <6 months Post Lung Dx | 7724 (39.8) | 3972 (33.3) | |
| | >6 months Post Lung Dx | 636 (3.3) | 276 (2.3) | |
| Any Oral Agent | N (%) Yes | 905 (4.7) | 294 (2.5) | <0.001 |
| 1st Radiation | Never | 8244 (42.5) | 5663 (47.5) | <0.001 |
| | <6 months Post Lung Dx | 9699 (50.0) | 5484 (46.0) | |
| | >6 months Post Lung Dx | 1449 (7.5) | 775 (6.5) | |
| Oxygen | N (%) Yes | 3053 (15.7) | 2589 (21.7) | <0.001 |
| SABA | N (%) Yes | 5013 (25.9) | 5866 (49.2) | <0.001 |
| LABA | N (%) Yes | 2787 (14.4) | 4578 (38.4) | <0.001 |
| Short Acting Anticholinergic | N (%) Yes | 1249 (6.4) | 1878 (15.8) | <0.001 |
| Long Acting Anticholinergic | N (%) Yes | 2714 (14.0) | 5211 (43.7) | <0.001 |
| ICS | N (%) Yes | 1749 (9.0) | 2965 (24.9) | <0.001 |
| Any Inhaler | N (%) Yes | 6465 (33.3) | 7445 (62.5) | <0.001 |
| Influenza | N (%) Yes | 115 (0.6) | 134 (1.1) | <0.001 |
| Pneumonia | N (%) Yes | 6054 (31.2) | 5534 (46.4) | <0.001 |
| COPD exacerbation | N (%) Yes | 1336 (6.9) | 3618 (30.4) | <0.001 |
| Overall Survival | N (%) Deaths | 17222 (88.8) | 10779 (90.4) | <0.001 |
| | Median (95% CI) Months | 5.3 (5.2, 5.5) | 4.3 (4.1, 4.5) | |
| | 1-year (95% CI) OS | 29.8% (29.1, 30.4) | 26.3% (25.5, 27.1) | |

although we cannot determine whether this radiation was of definitive or palliative intent. Data from the Netherlands has shown a similar decrease in surgery and increase in the use of radiation in the presence of comorbidity, especially COPD [12]. While previous data has shown seemingly low rates of adjuvant chemotherapy among individuals with resected lung cancer in Canada [22, 23], our study showed that individuals with COPD were somewhat less likely to receive adjuvant chemotherapy. Gao et al. reported worse disease-free survival in their meta-analysis, although it was not possible to discern whether this was due to less aggressive curative therapy or differences in stage [8]. Our data cannot reveal whether improvements in pre-habilitation or medical management would improve surgical rates and outcomes in the COPD population. However, there is some evidence that among individuals with airflow limitation who are able to undergo surgery, survival may not be impaired [24].

In those with more advanced disease, a worse survival among individuals with COPD was also noted. In our data set, the COPD population was less likely to receive systemic treatment on multivariable analysis, most of which would have been for palliative purposes in this population. While performance status affects the likelihood of receiving systemic treatment, it was not available in the data set, and so any association between COPD and poorer performance status is unclear. Nevertheless, COPD related frailty may have impacted treatment choice, and COPD severity itself has been shown to be related to mortality risk [6].

Not unexpectedly, a higher Charlson morbidity score conferred lower likelihood of receiving cancer directed therapies and worsened survival. Conversely, a higher ACG Resource Utilization band increased the likelihood of surgery and chemotherapy and slightly improved

**Table 4. Multivariable analysis of management and outcomes.**

| Covariate | Prognostic Factors for Surgery* (n = 104 843) Odds Ratio (95% CI), P value | Prognostic Factors for Chemotherapy* (n = 104 843) Odds Ratio (95% CI), P value | Prognostic Factors for Overall Survival (n = 104 843) Hazard Ratio (95% CI), P value |
|---|---|---|---|
| Year of Lung CA Diagnosis (Continuous) | 0.96 (0.96, 0.97), p<0.001 | 1.03 (1.02, 1.03), p<0.001 | 0.98 (0.98, 0.99), p<0.001 |
| Age (/ 5-year intervals) | 0.97 (0.96, 0.97), p<0.001 | 0.94 (0.94, 0.94), p<0.001 | 1.02 (1.02, 1.02), p<0.001 |
| Sex (F vs. M) | 1.15 (1.11, 1.19), p<0.001 | 0.98 (0.95, 1.01), p = 0.17 | 0.85 (0.84, 0.86), p<0.001 |
| Diagnosis of COPD on day of Lung Cancer dx vs not | 0.85 (0.81, 0.88), p<0.001 | 0.60 (0.58, 0.61), p<0.001 | 1.20 (1.19, 1.22), p<0.001 |
| Charlson Score (≥1 vs 0) | 0.59 (0.56, 0.63), <0.001 | 0.61 (0.58, 0.64), p<0.001 | 1.22 (1.20, 1.25), p<0.001 |
| Resource Utilization Bands (/ unit increase) | 1.24 (1.22, 1.27), p<0.001 | 1.04 (1.03, 1.06), p<0.001 | 0.95 (0.94, 0.96), p<0.001 |
| Income Quintile (/ quintile) | 1.07 (1.06, 1.09), p<0.001 | 1.09 (1.07, 1.09), p<0.001 | 0.97 (0.96, 0.97), p<0.001 |
| Rurality (No vs Yes) | 1.10 (1.05, 1.16), p<0.001 | 1.08 (1.04, 1.12), p<0.001 | 0.95 (0.94, 0.97), p<0.001 |
| Stage 1 | Reference | REFERENCE | REFERENCE |
| 2 | 0.82 (0.76, 0.89) | 7.70 (6.92, 8.58) | 1.77 (1.67, 1.87) |
| 3 | 0.09 (0.09, 0.10) | 12.45 (11.37, 13.64) | 3.48 (3.33, 3.63) |
| 4 | 0.01 (0.01, 0.01) | 6.24 (5.73, 6.79) | 7.33 (7.05, 7.61) |
| Unknown | 0.09 (0.08, 0.09) | 6.04 (5.53, 6.59) | 3.93 (3.78, 4.09) |
| | p<0.001 | p<0.001 | p<0.001 |

*Within 6 months of lung cancer diagnosis.

survival. This is likely because the ACG system is primarily designed to predict healthcare resource use [25]. It incorporates outpatient medical records (physician billing in our study) in addition to the hospital records employed by the Charlson score and then categorizes diseases by severity, chronicity, and types of resources required and generates groups predicting need.

The COPD group may have more significant respiratory care needs. Higher rates of hospitalization for respiratory infection were seen in our COPD population. This is consistent with previous data showing an increase in both frequency and duration of hospitalization in the COPD population compared with controls [26]. The COPD group more commonly received home oxygen and any medicated inhaler. Notably, higher cancer stage was inversely associated with inhaler use (70.9% in stage I/II vs. 62.5% in stage III/IV among those with COPD, p<0.001 on multivariate analysis, not shown), suggesting undertreatment of COPD. Despite these findings, we did not have data on the use of specialized palliative services, which is known to be reduced compared with lung cancer [27, 28]. Others have found a higher respiratory symptom burden in patients with COPD and lung cancer, supporting the need for optimal diagnosis and management [10].

Limitations to this study include a lack of histology data. The primary concern in this regard would be the contribution of small cell lung cancer (SCLC), as localized SCLC rarely involves surgery. Interestingly, two prior studies suggest SCLC is an under-represented histology in populations having COPD and lung cancer, at 6.5 and 7.8%, as compared with more commonly reported rates of 15% or higher for the overall lung cancer population [29–31]. If anything, the SCLC population would decrease the apparent surgical rate in our non-COPD population. In addition, the administrative data does not provide smoking history or specific spirometry values, both of which could alter likelihood of treatment.

We have already noted that our administrative data set could not provide performance status, which does effect likelihood of treatment. However, as regards the accuracy of patient-reported versus administrative database derived comorbidity, a Danish study determined that

only the latter could predict mortality, and that administrative data also reported more comorbidity [32]. In addition, due to the single payer public system within the province of Ontario, demographic data and records of surgical and radiation use are expected to be comprehensive. Intravenous chemotherapy is publicly funded for all, although universal public funding of oral cancer treatments and inhalers is limited to those 65 years and over.

COPD and lung cancer frequently coexist and our study demonstrates that a duel diagnosis is associated with poorer survival in both early and late stage lung cancer. This may be related to less frequent use of curative surgery as well as systemic treatments, along with higher rates of hospitalization for respiratory illness. Certainly, having both diseases increases the complexity of management, and impaired lung function is a major contraindication to surgery among those who are otherwise eligible [33]. While little evidence suggests that treatment of COPD will alter survival, improved care may help to decrease exacerbations and the symptom burden of COPD [34]. High quality, prospective research is required to determine whether the early detection and treatment of COPD is associated with improved outcomes in lung cancer patients.

## Author Contributions

**Conceptualization:** John R. Goffin, Gregory R. Pond.

**Data curation:** John R. Goffin, Grace H. Tang, Gregory R. Pond.

**Formal analysis:** John R. Goffin, Sophie Corriveau, Grace H. Tang, Gregory R. Pond.

**Investigation:** John R. Goffin, Sophie Corriveau, Grace H. Tang, Gregory R. Pond.

**Methodology:** John R. Goffin, Sophie Corriveau, Grace H. Tang, Gregory R. Pond.

**Project administration:** John R. Goffin, Gregory R. Pond.

**Software:** Gregory R. Pond.

**Supervision:** Gregory R. Pond.

**Validation:** John R. Goffin, Gregory R. Pond.

**Visualization:** Gregory R. Pond.

**Writing – original draft:** John R. Goffin, Gregory R. Pond.

**Writing – review & editing:** John R. Goffin, Sophie Corriveau, Grace H. Tang, Gregory R. Pond.

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
