## [Decision Letter · Decision Letter 0]

10 Mar 2021

PONE-D-21-01513

Management and Outcomes of Patients with Chronic Obstructive Lung Disease and Lung Cancer in a Public Healthcare System

PLOS ONE

Dear Dr. Goffin,

Thank you for submitting your manuscript to PLOS ONE. After careful consideration, we feel that it has merit but does not fully meet PLOS ONE’s publication criteria as it currently stands. Therefore, we invite you to submit a revised version of the manuscript that addresses the points raised during the review process. Both reviewers raised some minor points mainly related to statistcal and methodological issues. Please try to address them in the revised version.

We look forward to receiving your revised manuscript.

Kind regards,

Stelios Loukides

Academic Editor

PLOS ONE

Journal Requirements:

2. Please note that PLOS does not permit references to “data not shown.” Authors should provide the relevant data within the manuscript, the Supporting Information files, or in a public repository. If the data are not a core part of the research study being presented, we ask that authors remove any references to these data.

"Dr Goffin declares receiving Honorarium from Eisai (2020), Bristol-Myers Squibb (2020) and Merck (2018), conference travel support from AstraZeneca (2017), a speaking fee from Amgen (2018), and funding from the Canadian Partnership Against Cancer.  All other authors report no conflict of interest."

a. Please confirm that this does not alter your adherence to all PLOS ONE policies on sharing data and materials, by including the following statement: "This does not alter our adherence to  PLOS ONE policies on sharing data and materials.” (as detailed online in our guide for authors http://journals.plos.org/plosone/s/competing-interests).  If there are restrictions on sharing of data and/or materials, please state these.

Please note that we cannot proceed with consideration of your article until this information has been declared.

Reviewers' comments:

Reviewer's Responses to Questions

**Comments to the Author**

1. Is the manuscript technically sound, and do the data support the conclusions?

Reviewer #1: Yes

Reviewer #2: Yes

2. Has the statistical analysis been performed appropriately and rigorously? 

Reviewer #1: Yes

Reviewer #2: I Don't Know

3. Have the authors made all data underlying the findings in their manuscript fully available?

Reviewer #1: Yes

Reviewer #2: No

4. Is the manuscript presented in an intelligible fashion and written in standard English?

Reviewer #1: Yes

Reviewer #2: Yes

5. Review Comments to the Author

Reviewer #1: COPD is one of the top five major causes of morbidity and mortality worldwide [1]. Lung cancer remains one of the most fatal cancers, both due to delay in diagnosis and lack of effective treatments. COPD is an independent risk factor for lung carcinoma and lung cancer is up to five times more likely to occur in smokers with airflow obstruction than those with normal lung function [2]. This retrospective study shows that the presence of COPD in lung cancer patients is a major comorbidity, responsible for non-eligibility for appropriate therapeutic options, including surgery in early stage and palliative treatment in advanced stages.

There are some aspects of the study that need to be better clarified:

A. Diagnosis of COPD was made according to GOLD guidelines?

B. There is missing data for FEV1 (COPD stage) and the presence of emhysema. COPD patients were undertreated for lung cancer, irrespective of parameters such as FEV1 and presence and/or extent of emphysema?

C. There is missing data for smoking status and how current or ex smoking could affect management and outcome in lung cancer patients.

D. There is missing data for histologic subtype of lung cancer (adenoarcinoma, squamous carcinoma, SCLC) and how different histologic types might affect outcome, after adjusting for presence and/or severity of COPD.

E. Table 3: 17222 deaths in non-COPD lung cancer patients vs 10779 deaths in lung cancer patients without COPD. Is this correct, or the numbers have been put in reverse, in wrong columns?

F. PS is available only in 12.8% of patients. The results remain the same (less curative, less palliative treatment and poorer outcomes in lung cancer plus COPD) after adjusting according to PS?

G. The last paragraph of the discussion should be modified and presented in the following way:

"COPD and lung cancer frequently coexist and our study demonstrates that a duel diagnosis is associated with poorer survival in both early and late stage lung cancer. This may be related to less frequent use of curative surgery as well as systemic treatments, along with higher rates of hospitalization for respiratory illness. Both diseases create management difficulties for one another, but therapy for lung cancer is more crucially affected. Impaired lung function is the major cause of intolerance for surgical resection in an otherwise operative malignancy in stages I/II [3]. While little evidence suggests that treatment of COPD will alter survival, improved care may help to decrease exacerbations and the symptom burden of COPD(31).Further, better powered studies are required in order to clarify if early detection and treatment of COPD is associated with improved outcomes in lung cancer patients, especially in early stages of the disease.

The writers should consider the following sources:

[1]. . Rabe KF, Hurd S, Anzueto A, et al. Global strategy for the diagnosis, management and prevention of chronic obstructive pulmonary disease: GOLD executive summary. Am J Resp Crit Care Med. 2007;176(6):

532–555

[2]. Durham AL, Adcock IM. The relationship between COPD and lung cancer. Lung Cancer. 2015;90(2):121-127.

[3]. Hillas G, Perlikos F, Tsiligianni I, Tzanakis N. Managing comorbidities in COPD. Int J Chron Obstruct Pulmon Dis. 2015;10:95-109.

Reviewer #2: This is a well written manuscript that reports new insights on an important topic.

I have only a few minor suggestions for improvement, as follows.

Abstract

It’s not clear that ‘eligible’ refers to people with lung cancer.

I assume 05% is a typo for confidence intervals (replace with 95%).

Background

It’s not clear why, in the first instance, the hazard ratio given for people with COPD having lung cancer should adjust for smoking history since the main point is that both commonly co-occur because they are both smoking-related diseases.

Similarly, I was a little confused by the statement that ‘COPD is associated with higher mortality in the general population’; isn’t the main point that COPD and LC are both life-limiting illnesses?

I wasn’t sure what uses of chemotherapy and radiotherapy among those with COPD were being referred to, given that in the Abstract this seems to refer to both curative and palliative.

Methods

Did analyses examine the influence of being diagnosed with LC or COPD first or the time between diagnoses? And is there any evidence that diagnosis might have been delayed?

Discussion

I think the first paragraph unnecessarily repeats the aim and rationale of the study and is redundant.

In the limitations section, should it be acknowledged that the stage II/IV sample was not independent from the stage I/II? I wasn’t clear if the cohorts were completely separate or overlapping if followed over time.

It should also be noted that information on access to specialist palliative care was not available, and that this has been shown to be reduced for people with COPD compared to lung cancer by other studies.

I also think the fact the COPD group had lower income and were more likely to live in a rural location should be considered as a potential confounding factor for access to treatment (notwithstanding universal healthcare).

6. PLOS authors have the option to publish the peer review history of their article (what does this mean?). If published, this will include your full peer review and any attached files.

Reviewer #1: No

Reviewer #2: No

---

## [Author Response · Author response to Decision Letter 0]

22 Apr 2021

Editorial Board

PLOS ONE

RE: PONE-D-21-01513

Dear Editorial Board Member and Reviewers,

We are pleased to submit to you the revision of our manuscript, “Management and Outcomes of Patients with Chronic Obstructive Lung Disease and Lung Cancer in a Public Healthcare System.” Each Reviewer comment has been addressed below. We appreciate all of the feedback. 

Thank you again for considering our manuscript,

Sincerely,

John Goffin MD FRCPC

Editor’s Comments:

Please note that PLOS does not permit references to “data not shown.”

Response: Our sentence in the third paragraph of the Results regarding the stage III/IV population read “Similar differences were seen in the population age 65 years and over (data not shown).” Age was included in the multivariable analysis, making this statement redundant. It has been deleted.

We note that the grant information you provided in the ‘Funding Information’ and ‘Financial Disclosure’ sections do not match. When you resubmit, please ensure that you provide the correct grant numbers for the awards you received for your study in the ‘Funding Information’ section.

Response: This apparent mismatch may be because we were trying to tailor our responses to the available space. No actual grant or funding was received for this project. ICES does require an acknowledgement of Ministry of Health funding of ICES itself, as below:

Support to ICES is provided from an annual grant by the Ministry of Health and Long-Term Care (MOHLTC) and the Ontario Institute for Cancer Research (OICR). The opinions, results and conclusions reported in this paper are those of the authors. No endorsement by ICES, Cancer Care Ontario, OICR or the Government of Ontario is intended or should be inferred. The funding source had no role in the collection, analysis, and interpretation of data and in writing the manuscript.

Thank you for stating the following in the Competing Interests section…

Response: The previous statement of conflict remains up to date, with required language added as below. 

Dr Goffin declares receiving Honorarium from Eisai (2020), Bristol-Myers Squibb (2020) and Merck (2018), conference travel support from AstraZeneca (2017), a speaking fee from Amgen (2018), and funding from the Canadian Partnership Against Cancer. All other authors report no conflict of interest. This does not alter our adherence to PLOS ONE policies on sharing data and materials. Data may be obtained from a third party and are not publicly available. A data request can be sent to ICES (formerly the Institute for Clinical Evaluative Sciences): https://www.ices.on.ca/About-ICES/ICES-Contacts-and-Sites/contact-form

We note that you have indicated that data from this study are available upon request. PLOS only allows data to be available upon request if there are legal or ethical restrictions on sharing data publicly.

Response: We did not find a flexible way of responding to the data access question, but in fact data is not available due to privacy and legal restrictions. ICES is subject to Ontario’s Personal Health Information Privacy Act and has data sharing agreements with health data sources and the Ministry of Health and Long-term Care of Ontario. For reasons of privacy in accessing patient level administrative healthcare information, data sets cannot be accessed outside of ICES servers, and can only be accessed by ICES approved researchers. The following excerpt is from the ICES website, https://www.ices.on.ca:

DAS staff work with requestors to design a research-ready data extract from the ICES data repository using an algorithm that de-identifies data in a way that provides researchers with individual-level data while meeting all privacy standards. Requestors are provided with access to the linked, de-identified data extract on a secure, online research environment, where they can perform analyses and create reports. To ensure privacy policy compliance, linked data cannot be copied or removed from the environment.

Reviewer #1: COPD is one of the top five major causes of morbidity and mortality worldwide [1]. Lung cancer remains one of the most fatal cancers, both due to delay in diagnosis and lack of effective treatments. COPD is an independent risk factor for lung carcinoma and lung cancer is up to five times more likely to occur in smokers with airflow obstruction than those with normal lung function [2]. This retrospective study shows that the presence of COPD in lung cancer patients is a major comorbidity, responsible for non-eligibility for appropriate therapeutic options, including surgery in early stage and palliative treatment in advanced stages.

There are some aspects of the study that need to be better clarified:

A. Diagnosis of COPD was made according to GOLD guidelines?

Response: Our data is derived from administrative data sets. Because spirometry values are not available, COPD was defined based on the validated work of Gershon et al. described in the Methods (Reference 21). Gershon et al. compared administrative data with a chart review to derive the administrative data associated with a COPD diagnosis. ICES (previously known as the Institute for Clinical Evaluative Sciences) uses this to generate the cohort of individuals having this definition of COPD. 

B. There is missing data for FEV1 (COPD stage) and the presence of emhysema. COPD patients were undertreated for lung cancer, irrespective of parameters such as FEV1 and presence and/or extent of emphysema?

Response: Unfortunately, as noted above, specific spirometry values are not available from the data set, so we used a defined COPD population. It is therefore true we cannot determine the association between FEV1 or emphysema severity and outcomes. This limitation has been added to the last paragraph of the Discussion beginning ‘Limitations to this study include…’.

In addition, the administrative data does not provide smoking history or specific spirometry values, both of which could alter likelihood of treatment.

C. There is missing data for smoking status and how current or ex smoking could affect management and outcome in lung cancer patients.

Response: Tobacco use by individual would be wonderful to have but it is not yet available from ICES. It is our hope that data collected from cancer centres will allow this within a couple of years. This is a limitation of administrative data. This limitation has been added to the last paragraph of the Discussion beginning ‘Limitations to this study include…’.

In addition, the administrative data does not provide smoking history or specific spirometry values, both of which could alter likelihood of treatment.

D. There is missing data for histologic subtype of lung cancer (adenoarcinoma, squamous carcinoma, SCLC) and how different histologic types might affect outcome, after adjusting for presence and/or severity of COPD.

Response: We do note lack of histology in our limitations, particularly as regards small-cell lung cancer. While this data would be helpful, the low frequency of SCLC in other COPD data sets suggests it would have a small impact on our analysis.

E. Table 3: 17222 deaths in non-COPD lung cancer patients vs 10779 deaths in lung cancer patients without COPD. Is this correct, or the numbers have been put in reverse, in wrong columns?

Response: The numbers as show in Table 3 are correct, but need to be considered in the context of the denominators (shown in the column header). Thus, there were 17222 deaths among 19392 persons without COPD (88.8%) and 10779 deaths among 11922 persons without COPD (90.4%). 

F. PS is available only in 12.8% of patients. The results remain the same (less curative, less palliative treatment and poorer outcomes in lung cancer plus COPD) after adjusting according to PS?

Response: We think the Reviewer is referring to the fact that 12.8% of individuals had a Charlson Score of ≥ 1, while the rest were at zero. The analysis did adjust for the Charlson Score. 

G. The last paragraph of the discussion should be modified and presented in the following way:

"COPD and lung cancer frequently coexist and our study demonstrates that a duel diagnosis is associated with poorer survival in both early and late stage lung cancer. This may be related to less frequent use of curative surgery as well as systemic treatments, along with higher rates of hospitalization for respiratory illness. Both diseases create management difficulties for one another, but therapy for lung cancer is more crucially affected. Impaired lung function is the major cause of intolerance for surgical resection in an otherwise operative malignancy in stages I/II [3]. While little evidence suggests that treatment of COPD will alter survival, improved care may help to decrease exacerbations and the symptom burden of COPD(31).Further, better powered studies are required in order to clarify if early detection and treatment of COPD is associated with improved outcomes in lung cancer patients, especially in early stages of the disease.

The writers should consider the following sources:

[1]. . Rabe KF, Hurd S, Anzueto A, et al. Global strategy for the diagnosis, management and prevention of chronic obstructive pulmonary disease: GOLD executive summary. Am J Resp Crit Care Med. 2007;176(6):

532–555

[2]. Durham AL, Adcock IM. The relationship between COPD and lung cancer. Lung Cancer. 2015;90(2):121-127.

[3]. Hillas G, Perlikos F, Tsiligianni I, Tzanakis N. Managing comorbidities in COPD. Int J Chron Obstruct Pulmon Dis. 2015;10:95-109.

Response: We agree these are good references. Our reference Vogelmeier et al., 2017, is a more recent version of Rabe et al., 2007, above, and we have a well-highlighted copy of Durham et al. on file. We have adjusted the concluding paragraph, adding the suggested reference of Hillas et al., as follows:

COPD and lung cancer frequently coexist and our study demonstrates that a duel diagnosis is associated with poorer survival in both early and late stage lung cancer. This may be related to less frequent use of curative surgery as well as systemic treatments, along with higher rates of hospitalization for respiratory illness. Certainly, having both diseases increases the complexity of management, and impaired lung function is a major contraindication to surgery among those who are otherwise eligible(33). While little evidence suggests that treatment of COPD will alter survival, improved care may help to decrease exacerbations and the symptom burden of COPD(34). High quality, prospective research is required to determine whether the early detection and treatment of COPD is associated with improved outcomes in lung cancer patients.

Reviewer #2: This is a well written manuscript that reports new insights on an important topic.

I have only a few minor suggestions for improvement, as follows.

Abstract

It’s not clear that ‘eligible’ refers to people with lung cancer.

Response: To clarify, the wording was changed from ‘eligible individuals’ to ‘individuals with lung cancer’.

I assume 05% is a typo for confidence intervals (replace with 95%).

Response: Thank you for catching this typo. It has been corrected.

Background

It’s not clear why, in the first instance, the hazard ratio given for people with COPD having lung cancer should adjust for smoking history since the main point is that both commonly co-occur because they are both smoking-related diseases.

Response: It is true that both COPD and lung cancer are highly associated with tobacco use. It is also true that not all COPD is the product of tobacco use (alpha-1 antitrypsin deficiency, likely air pollution) and a growing minority of lung cancer is not induced by tobacco. Adjusting for tobacco use thus supports an independent association between COPD and lung cancer.

Similarly, I was a little confused by the statement that ‘COPD is associated with higher mortality in the general population’; isn’t the main point that COPD and LC are both life-limiting illnesses?

Response: Agreed, both COPD and lung cancer are life-limiting. In the introduction, we are trying to put the current paper in context. Our paper shows that prognosis is worse and fewer treatments are undertaken in those who have COPD in addition to lung cancer. Acknowledging the underlying risk of death from COPD adds plausibility to the notion that a combination of these two diagnoses could be problematic.

I wasn’t sure what uses of chemotherapy and radiotherapy among those with COPD were being referred to, given that in the Abstract this seems to refer to both curative and palliative.

Response: In the Results section, in the paragraph starting ‘Among persons with stage I/II disease…,’ we note decreased use of adjuvant chemotherapy and increased use of radiotherapy. Most of this radiotherapy is likely to be of curative intent (in lieu of surgery), but we cannot know what portion might have been palliative. We note this uncertainty in the Discussion.

In the next Results paragraph starting ‘In the population with stage III/IV disease,’ we note less use of both treatments. We cannot discern from administrative data if some patients with stage III disease could not receive curative radiation plus chemotherapy due to comorbidity etc. or due to extent of disease (eg. stage IIIc). To clarify this uncertainty, a sentence in the fourth paragraph of the Discussion was altered (underlined phrase):

In our data set, the COPD population was less likely to receive systemic treatment on multivariable analysis, most of which would have been for palliative purposes in this population.

Methods

Did analyses examine the influence of being diagnosed with LC or COPD first or the time between diagnoses? And is there any evidence that diagnosis might have been delayed?

Response: Our prior paper showed that a large portion of individuals with stage III-IV lung cancer never undergo spirometry (45.6%), and undoubtedly there is underdiagnosis and delayed diagnosis [Corriveau et al. BMC Cancer 2021 21:14]. However, our analysis does not include post lung cancer diagnoses of COPD, and so we cannot discern what impact that might have on treatment. 

Discussion

I think the first paragraph unnecessarily repeats the aim and rationale of the study and is redundant.

Response: This is a fair comment. It does not seem necessary for flow and has been deleted.

In the limitations section, should it be acknowledged that the stage II/IV sample was not independent from the stage I/II? I wasn’t clear if the cohorts were completely separate or overlapping if followed over time.

Response: The stage I/II and III/IV cohorts were designed to be independent. Stage is only entered once in the cancer registry. In the stage I/II curative population, only treatments undertaken within 6 months of diagnosis were registered. During this window, very few progressing, metastatic patients should contaminate this group once formal staging is complete and recorded. The following statement has been added to the Methods (end of paragraph two):

Early stage (I/II) and late stage (III/IV) cancer treatments were designed to be independent, with treatment after early stage treatment being limited to surgery within 3 months of diagnosis or chemotherapy or radiation initiated within 6 months. 

It should also be noted that information on access to specialist palliative care was not available, and that this has been shown to be reduced for people with COPD compared to lung cancer by other studies.

Response: Agreed. We have added a sentence to the Discussion paragraph beginning ‘The COPD group may have more significant respiratory care needs’ with two relevant references:

Despite these findings, we did not have data on the use of specialized palliative services, which is known to be reduced compared with lung cancer(27,28).

I also think the fact the COPD group had lower income and were more likely to live in a rural location should be considered as a potential confounding factor for access to treatment (notwithstanding universal healthcare).

Response: Income and rural habitation were assessed in the multivariable analysis. They did impact the likelihood of treatment, but COPD was independently predictive. This is noted in second to last paragraph of the Results.

---

## [Decision Letter · Decision Letter 1]

5 May 2021

Management and outcomes of patients with chronic obstructive lung disease and lung cancer in a public healthcare system

PONE-D-21-01513R1

Dear Dr. Goffin,

We’re pleased to inform you that your manuscript has been judged scientifically suitable for publication and will be formally accepted for publication once it meets all outstanding technical requirements.

Kind regards,

Stelios Loukides

Academic Editor

PLOS ONE

Additional Editor Comments (optional):

Reviewers' comments:

Reviewer's Responses to Questions

**Comments to the Author**

1. If the authors have adequately addressed your comments raised in a previous round of review and you feel that this manuscript is now acceptable for publication, you may indicate that here to bypass the “Comments to the Author” section, enter your conflict of interest statement in the “Confidential to Editor” section, and submit your "Accept" recommendation.

Reviewer #1: All comments have been addressed

Reviewer #2: All comments have been addressed

2. Is the manuscript technically sound, and do the data support the conclusions?

Reviewer #1: Yes

Reviewer #2: Yes

3. Has the statistical analysis been performed appropriately and rigorously? 

Reviewer #1: N/A

Reviewer #2: Yes

4. Have the authors made all data underlying the findings in their manuscript fully available?

Reviewer #1: Yes

Reviewer #2: Yes

5. Is the manuscript presented in an intelligible fashion and written in standard English?

Reviewer #1: Yes

Reviewer #2: Yes

6. Review Comments to the Author

Reviewer #1: This is a well-written paper highlighting that, co-existence of COPD and lung cancer create management difficulties for both diseases, with the treatment of lung cancer being more crucially affected. Impaired lung function is a major contraindication for surgical approach in an otherwise operative malignancy in early stages and increases the intolerance in systemic and palliative treatment in advanced stages. The authors provide satisfactory explanations in all questions that were posed.I have no further comments.

Reviewer #2: (No Response)

7. PLOS authors have the option to publish the peer review history of their article (what does this mean?). If published, this will include your full peer review and any attached files.

Reviewer #1: No

Reviewer #2: No

---

## [Editor Report · Acceptance letter]

7 May 2021

PONE-D-21-01513R1 

Management and outcomes of patients with chronic obstructive lung disease and lung cancer in a public healthcare system 

Dear Dr. Goffin:

I'm pleased to inform you that your manuscript has been deemed suitable for publication in PLOS ONE. Congratulations! Your manuscript is now with our production department. 

Kind regards, 

on behalf of

Dr. Stelios Loukides 

Academic Editor

PLOS ONE